# Establishment and Characterization of Behavioral Changes in the Nuclear Localization Human α-Synuclein Transgenic Mice

**DOI:** 10.3390/diseases13080261

**Published:** 2025-08-14

**Authors:** Ziou Wang, Mengchen Wei, Shengtao Fan, Zheli Li, Weihu Long, Haiting Wu, Yiwei Zhang, Zhangqiong Huang

**Affiliations:** Institute of Medical Biology, Chinese Academy of Medical Sciences and Peking Union Medical College, No. 935, Jiaoling Road, Kunming 650118, China; wzo@pumc.edu.cn (Z.W.); weimch@imbcams.com.cn (M.W.); fst@imbcams.com.cn (S.F.); lzl@imbcams.com.cn (Z.L.); longweihu@imbcams.com.cn (W.L.); wuht@student.pumc.edu.cn (H.W.); zyw@student.pumc.edu.cn (Y.Z.)

**Keywords:** human α-synuclein, transgenic mice, nuclear localization, motor dysfunction, anxiety-like behavior

## Abstract

**Objectives:** This study aimed to establish a transgenic mouse model expressing nucleus-localized human α-synuclein (α-syn) to investigate its impact on the central nervous system and behavior and the underlying mechanisms involved. **Methods:** A nuclear localization sequence (NLS) was added to the end of the human SNCA (hSNCA) gene. Subsequently, an empty vector and a mammalian lentiviral vector of the hSNCA-NLS were constructed. Transgenic mice were generated via microinjection, with genotyping and protein expression confirmed by PCR and western blotting. Only male mice were used in subsequent behavioral and molecular experiments. Immunofluorescence identified the colocalization of human α-syn with the cell nucleus in mouse brain tissues. Behavioral changes in transgenic mice were assessed using open field, rotarod, and O-maze tests. qPCR and Western blotting detected expression levels of genes and proteins related to inflammation, endoplasmic reticulum stress (ERS), and apoptosis. Bulk RNA sequencing was used to screen for differentially expressed genes and signaling pathways. **Results:** We successfully constructed a transgenic mouse model expressing human α-syn. Human α-syn was widely expressed in the heart, liver, spleen, kidneys, and brain of the mice, with distinct nuclear localization observed. Behavioral assessments demonstrated that, by 2 months of age, the mice exhibited motor dysfunction alongside astrocyte proliferation and neuroinflammation. At 6 months, the elevated expression of ERS-related genes (ATF6, PERK, and IRE1) and activation of the PERK-Beclin1-LC3II pathway indicated progressive ERS. By 9 months, apoptotic events had occurred, accompanied by significant anxiety-like behaviors. Bulk RNA sequencing further identified key differentially expressed genes, including IL-1α, TNF, PERK, BECLIN, GABA, IL-6α, P53, LC3II, NOS, and SPAG, suggesting their involvement in the observed pathological and behavioral phenotypes. **Conclusions:** The nuclear localization human α-syn transgenic mice were successfully established. These findings demonstrate that nucleus-localized α-syn induces early motor deficits, which are likely mediated by neuroinflammation, whereas later anxiety-like behaviors may result from ERS-induced apoptosis. This model provides a valuable tool for elucidating the role of nuclear α-syn in Parkinson’s disease and supports further mechanistic and therapeutic research.

## 1. Introduction

Parkinson’s disease (PD) represents the second most prevalent neurodegenerative disorder following Alzheimer’s disease, affecting over 6 million individuals worldwide. This prevalence reflects a 2.5-fold increase over the past three decades, positioning PD as a major contributor to neurological disability [1]. The disease is pathologically characterized by the progressive and selective degeneration of dopaminergic neurons in the substantia nigra pars compacta, which leads to the significant depletion of striatal dopamine and the formation of Lewy bodies in the surviving neurons [2]. α-Syn, the principal protein component of Lewy bodies, is central to PD pathogenesis; its abnormal overexpression and aggregation into higher-order pathological structures drive neurodegenerative processes. These molecular mechanisms have established PD as a type of synucleinopathy and identified α-syn as its definitive biomarker [3].

α-Syn is a soluble small-molecular-weight protein abundantly expressed in presynaptic and perinuclear regions of the central nervous system, accounting for approximately 1% of total neuronal cytoplasmic proteins [4]. While most previous studies have focused on cytoplasmic α-syn due to its colocalization with Lewy bodies, emerging evidence has indicated that under specific pathological conditions, α-syn exhibits a distinct nuclear localization preference and tends to aggregate within the nucleus [5]. Notably, α-syn-positive nuclear inclusions have been detected in the pontine base neurons and oligodendroglial nuclei of multiple system atrophy patients [6] and in postmortem tissues from individuals with Lewy body dementia, where prominent nuclear accumulation of α-syn was consistently observed [7]. Both PD patient brains and oxidative stress-induced cellular models demonstrate a marked propensity for α-syn accumulation within the nuclear compartment, with these aberrant nuclear aggregates exhibiting direct correlations with cytotoxicity [8,9]. Mechanistic studies have revealed that the nuclear localization of α-syn is strongly dependent on its phosphorylation at serine 129 (S129), a well-established pathological hallmark of PD [10,11], suggesting that nucleus-localized α-syn may exert its pathogenic effects through this specific posttranslational modification. Transcriptional profiling further revealed elevated expression of nucleus-localized α-syn, particularly within neuronal populations vulnerable to PD pathology, reinforcing its potential role in disease-specific neurodegeneration [12]. A recent study on α-syn nuclear import in transgenic mice revealed that the nuclear localization of α-syn was associated with PD-like motor dysfunction and gastrointestinal alterations [13]. Intriguingly, α-syn demonstrates preferential nuclear expression in specific cell types, particularly during the terminal differentiation of human fibroblasts and erythroid cells [4]. Furthermore, accumulating evidence has consistently revealed nuclear α-syn localization across diverse experimental systems, including cultured SH-SY5Y and MES23.5 dopaminergic cell lines [14,15], mouse primary cortical neurons [16], dopaminergic neuronal precursor cell lines [10], normal rat bone marrow [17], mouse brain tissue [18], and postmortem human brains [19], suggesting that this phenomenon represents a fundamental biological characteristic rather than being limited to pathological conditions.

The nuclear localization of α-syn suggests that it may have functional significance in the nuclear compartment and a close association with the pathogenesis of neurodegenerative disorders, although the precise mechanisms remain elusive. In our preliminary research, we successfully engineered a recombinant adeno-associated viral vector (pAAV-IRES-hrGFP-α-Syn-3*NLS) and established a whole-brain overexpression model of nuclear-targeted human α-syn through intracerebroventricular injection. While this transgenic mouse model offers the advantage of rapid onset of pathology in specific brain regions following localized injection, it is limited by transient expression and an inability to achieve systemic expression (manuscript in preparation). The pAAV-IRES-hrGFP-α-Syn-3*NLS transgenic mouse model has inherent limitations in investigating α-syn aggregation-related pathologies beyond the nervous system. To address this issue, we developed an improved model system by constructing lentiviral vectors expressing either hSNCA-NLS (experimental group) or EGFP (control group). These vectors were subsequently used to generate α-syn transgenic mice with nuclear localization via microinjection. This advanced model enables the systematic exploration of behavioral alterations induced by nuclear-targeted α-syn and facilitates the mechanistic elucidation of its role in PD-like neurodegeneration. The establishment of this model enhances our understanding of disease pathogenesis and provides a valuable platform for developing novel therapeutic strategies and pharmacological interventions for PD patients.

## 2. Materials and Methods

### 2.1. Ethics Statement

All experimental mice were provided by the Laboratory Animal Department of the Institute of Medical Biology, Chinese Academy of Medical Sciences [Animal Production License No. SCXK (Dian) K2022-0002]. The animals were housed in a barrier facility maintained at 20–24 °C with ad libitum access to food and water under a 12 h light/dark cycle [Animal Facility License No. SYXK (Dian) K2022-0006]. The experimental procedures were approved by the Institutional Animal Care and Use Committee (IACUC) of the Institute of Medical Biology, Chinese Academy of Medical Sciences (Approval No. DWSP202209006) and strictly adhered to the 3R principles (replacement, reduction, and refinement) of humane animal research.

### 2.2. Lentiviral Vector Construction and Establishment of Human α-Syn Transgenic Mice

The lentiviral vectors pLV-hSNCA-NLS and pLV-EGFP were constructed by VectorBuilder Inc. (Guangzhou, China). Female ICR mice (7–8 weeks old) served as pseudopregnant recipients, whereas female C57BL/6 mice (7–8 weeks old) were used as donors. Superovulation was induced by the intraperitoneal injection of 10 IU PMSG (day 1), followed by hCG administration 48 h later. After mating superovulated C57BL/6 females with C57BL/6 males, fertilized zygotes were collected from the oviducts upon identification of the vaginal plugs. At 24–25 h post hCG, pronuclear-stage zygotes were microinjected with pLV-hSNCA-NLS or pLV-EGFP (100–200 ng/μL) and cultured at 37 °C under 5% CO_2_. Embryos that reached the 2-cell stage were surgically transferred into the oviducts of 0.5 dpc pseudopregnant ICR females. Recipient mice were maintained in a barrier facility post recovery, with thermal support provided until they were fully ambulatory.

### 2.3. PCR

F0-generation male mice were bred with wild-type C57BL/6 females to generate F1 offspring, which were subsequently screened for transgene integration. Surgical scissors and forceps were sterilized with 75% ethanol before tissue collection. Approximately 0.5 cm of the tail tip was excised and digested in 100 μL of lysis buffer at 55 °C for 15 min, followed by heat inactivation at 95 °C for 5 min. The reaction was terminated with 100 μL of Stop Solution, followed by vortex mixing. PCR amplification was performed in a 20 μL reaction mixture containing 8 μL of ddH2O, 1 μL of primer mixture (10 μM each), 1 μL of template DNA, and 10 μL of PCR Master Mix. The thermal cycling conditions included initial denaturation at 94 °C for 3 min; 35 cycles of 94 °C for 30 s, 60 °C for 30 s, and 72 °C for 20 s; and a final extension at 72 °C for 10 min.

### 2.4. Western Blotting

Brain tissue homogenization was performed via the use of zirconium beads in RIPA lysis buffer, followed by a 20 min incubation on ice. The lysates were centrifuged at 12,000× *g* for 10 min at 4 °C, and the supernatants were quantified via a BCA protein assay kit. Protein concentrations were normalized to 3 μg/μL, mixed with 5× SDS–PAGE loading buffer, and denatured by boiling for 10 min. Proteins were resolved on SDS–polyacrylamide gels (80 V for 30 min then 120 V for 60 min) and transferred to PVDF membranes via a semidry transfer apparatus. The membranes were blocked with 5% BSA for 60 min at room temperature and then incubated with primary antibodies overnight at 4 °C. After three 10 min TBST washes, the membranes were probed with species-appropriate secondary antibodies for 60 min at room temperature. Following additional TBST washes (3 × 10 min), protein bands were visualized using an Odyssey infrared imaging system.

### 2.5. Immunofluorescence Staining

The mice were deeply anesthetized with 1.25% avertin (0.02 mL/g) and transcardially perfused with 4% paraformaldehyde (PFA). The brains were subsequently fixed in 4% PFA for 48 h, dehydrated, and embedded in paraffin for 4 μm sectioning. The sections were incubated at 65 °C, deparaffinized, rehydrated, and subjected to antigen retrieval in EDTA buffer via microwave heating for 25 min. After being blocked with goat serum (60 min, RT), the sections were incubated with primary antibodies at 4 °C overnight, washed with PBS (3 × 5 min), probed with species-matched fluorescent secondary antibodies, and counterstained with DAPI. Slides were imaged with a Pannoramic MIDI scanner.

### 2.6. Behavioral Tests

#### 2.6.1. Open Field Test

On the test day, the mice were acclimated to the experimental area for 2 h before testing. The open field apparatus was cleaned with 75% ethanol, and the parameters (animal ID, date, and experimental conditions) were configured in the tracking software. After complete ethanol evaporation, each mouse was gently transferred from its home cage to the center of the open field arena and allowed to explore freely for 5 min while being recorded by the automated behavioral tracking system. The arena was thoroughly cleaned with 75% ethanol to eliminate odor cues between trials, and testing proceeded only after complete drying.

#### 2.6.2. O-Maze Test

On the test day, the mice were acclimated to the experimental room for 2 h before the procedure. The maze apparatus was thoroughly cleaned with 75% ethanol, and all experimental parameters (animal ID, date, and trial conditions) were preconfigured in the behavioral tracking software. After complete ethanol evaporation, each mouse was gently placed at the junction between an open and closed quadrant of the maze to initiate exploration. The 5 min test session was automatically recorded via video tracking software to quantify anxiety-related behaviors. Between trials, the maze was meticulously cleaned with 75% ethanol to eliminate olfactory cues, with subsequent testing commencing only after complete drying.

#### 2.6.3. Rotarod Test

On the test day, the mice were acclimated to the experimental environment for 2 h. One hour before formal testing, each mouse received a 5 min training session on the rotarod apparatus to establish baseline performance. The rod was cleaned with 75% ethanol before testing, with experiments commencing after complete evaporation. The protocol consisted of a 5 min trial: an initial 2 min acceleration phase (5–40 rpm) followed by a 3 min constant speed phase (40 rpm). Each mouse underwent three trials with 30 min intertrial intervals, and the mean latency to fall was calculated. Between cohorts, the rod was thoroughly cleaned with 75% ethanol and allowed to dry completely.

### 2.7. Real-Time PCR

The tissues were homogenized in Trizol (1 mL) and incubated on ice (10 min). After centrifugation (12,000× *g*, 2 min, 4 °C), 800 μL of the supernatant was mixed with chloroform (200 μL) and centrifuged (12,000× *g*, 15 min, 4 °C). The aqueous phase (400 μL) was precipitated with isopropanol (600 μL, 10 min RT), centrifuged (12,000× *g*, 10 min, 4 °C), and washed twice with 75% ethanol (1 mL). The RNA pellets were air dried (15 min), dissolved in RNase-free water (20 μL), and normalized to 1000 ng/μL. cDNA was synthesized via a reverse transcription kit (25 °C/5 min → 42 °C/60 min → 70 °C/15 min). qPCR was performed in 20 μL reactions (95 °C/5 min initial denaturation, followed by 40 cycles of 95 °C/15 s and 60 °C/60 s) via SYBR Green chemistry. The data were analyzed via the 2^−ΔΔCt^ method.

### 2.8. Bulk RNA and Whole-Genome Sequencing with Bioinformatics Analysis

Brain tissues from three 12-month-old male transgenic mice (confirmed positive for nucleus-localized human α-syn by PCR and Western blot) were submitted to Shanghai Biotechnology Corporation for whole-genome sequencing and bulk RNA sequencing (bulk RNA-seq). The genomic sequencing data were processed via Tablet (https://ics.hutton.ac.uk/tablet/, accessed on 29 February 2024), with the inserted sequences aligned and annotated via SnapGene 4.3.6. For bulk RNA-seq analysis, differentially expressed genes (DEGs) were functionally characterized via DAVID Bioinformatics Resources 6.8 (https://david.ncifcrf.gov/, accessed on 13 March 2024). for Gene Ontology (GO) terms (biological process/BP, cellular component/CC, and molecular function/MF) and Kyoto Encyclopedia of Genes and Genomes (KEGG) pathways. The enrichment results were visualized as bubble plots via the Bioinformatics online platform (https://www.bioinformatics.com.cn/, accessed on 13 March 2024). Apoptosis-related pathways were further investigated using the KEGG database (https://www.kegg.jp/kegg/pathway.html, accessed on 13 March 2024).

### 2.9. Statistical Analysis

All behavioral data were acquired via the SMART V3.0 video tracking system and are expressed as the means ± standard deviations (SDs). Statistical analyses were performed via GraphPad Prism 8.0 software, with group comparisons assessed via unpaired Student’s *t* tests. Differences were considered statistically significant at *p* < 0.05 (*, **, ***, and **** denote *p* < 0.05, <0.01, <0.001, and <0.0001, respectively). Figures were processed and arranged via Adobe Illustrator 2020.

## 3. Results

### 3.1. Construction of Mammalian Gene Expression Lentiviral Vectors

Given that lentiviral vectors can integrate into the host genome and enable stable long-term gene expression [20] and that the EF1A promoter has the highest stability in embryonic stem cells [21], the lentiviral system was selected as the delivery vehicle, with EF1A as the promoter and human SNCA cDNA as the target gene for expression. The designed hSNCA-NLS plasmid and the EGFP empty plasmid were sent to VectorBuilder Inc. (Chicago, IL, USA) for the construction and packaging of the mammalian gene expression lentiviral vectors pLV-hSNCA-NLS and pLV-EGFP (Figure 1).

### 3.2. Generation of Human α-Syn Transgenic Mice

A total of 30 female mice were superovulated, yielding 1010 embryos, of which 830 fertilized eggs were microinjected. The embryos were subsequently transplanted into 25 recipient mice (30 embryos per mouse), resulting in 16 pregnancies (64% success rate) and 51 pups. PCR genotyping of tail-derived DNA identified 29 positive founder mice (F0; 18 males and 11 females), with a 40.8% positive rate (Figure 2A). Male founders F0-1 and F0-2 were bred with wild-type C57BL/6 females, and their F1 offspring were genotyped at 14 days postpartum. EGFP transgenic mice presented a distinct DNA band at 232 bp, whereas human α-synuclein transgenic mice presented two clear bands at 232 bp and 290 bp (Figure 2B), confirming successful genomic integration and stable germline transmission. To assess systemic expression, human α-syn protein levels were examined through the Western blot of multiple F1 tissues, including the heart, liver, spleen, kidney, and brain, revealing widespread expression (Figure 2C). As shown in the figure, the expression of hα-syn was validated through PCR and Western blot analysis in three mice per group. Whole-genome sequencing of three F1-positive mice precisely mapped the hSNCA-NLS insertion site to chromosome 6 (Figure 2D), with the integrated sequence matching the designed construct (Figure 2E).

### 3.3. Human α-Syn Exhibited Distinct Colocalization with the Cell Nucleus

Brain tissue sections from the transgenic mice were subjected to immunofluorescence staining to further validate the successful nuclear localization in the transgenic mice. The results demonstrated a distinct colocalization of human α-syn with the cell nucleus (Figure 3). Collectively, the findings from PCR genotyping, Western blot protein identification, and immunofluorescence colocalization confirmed the successful construction of the nuclear localization human α-syn transgenic mouse model.

### 3.4. Nuclear Localization of Human α-Syn in Transgenic Mice Resulted in Motor Dysfunction and Anxiety-like Behaviors

Given that the primary clinical feature of PD patients was motor dysfunction [22] and that the open-field test was a behavioral experiment used to assess the locomotor ability of mice [23], we assigned 13 identified male nuclear localization human α-syn transgenic mice as the experimental group and 13 male EGFP mice as the control group. These groups were subjected to open-field tests (*p* < 0.001 or *p* < 0.0001) (Figure 4A,B) and rotarod tests (*p* < 0.05, *p* < 0.01 or *p* < 0.001) (Figure 4C) at 1, 2, 3, 4, 6, 9, and 12 months of age. The results indicated that the experimental group exhibited significant motor impairments as early as 2 months of age, which persisted until 12 months, demonstrating that the nuclear localization of human α-syn in the transgenic mice resulted in motor dysfunction at an earlier stage. Since 30–40% of PD patients clinically exhibited significant anxiety symptoms [24] and the prevalence of anxiety disorders in PD patients is significantly greater than that in the general population [25], we conducted an O maze test (*p* < 0.05, *p* < 0.01) (Figure 4D,E) at the same time points. The experimental group displayed anxiety-like behaviors starting at 9 months of age, which continued until 12 months. Considering the anxiety-like behaviors observed in the nuclear localization human α-syn transgenic mice during the behavioral test, we examined the expression of the GABA gene, a key emotional factor. The results revealed that the expression of the GABA gene in these transgenic mice significantly decreased at 9 months of age but remained significantly reduced until 12 months of age (*p* < 0.05, *p* < 0.01) (Figure 4F).

### 3.5. Brain Tissue of Nuclear Localization Human α-Syn Transgenic Mice Exhibited Astrocyte Proliferation and Inflammatory Responses

Given the close association between PD and astrocytes [26], we examined astrocyte activity in the brain tissues of both groups of male mice. In the nuclear localization of human α-syn in transgenic mice, the expression of glial fibrillary acidic protein (GFAP), a marker of astrocytes, began to significantly differ as early as 2 months of age, and this significant difference persisted until 12 months of age (*p* < 0.05, *p* < 0.01 or *p* < 0.001) (Figure 5A,B). Under pathological conditions, astrocytes have been shown to produce inflammatory cytokines. Therefore, we further investigated whether an inflammatory response had occurred in the brain tissues of these mice. The results revealed that IFN-γ and NF-κB levels in the brain tissues of nuclear localization human α-syn transgenic mice were already significantly different at 1 month of age, whereas levels of other inflammatory cytokines began to diverge at 2 months and remained altered through 12 months. (*p* < 0.05, *p* < 0.01, *p* < 0.001 or *p* < 0.0001) (Figure 5C).

### 3.6. Nuclear Localization Human α-Syn Transgenic Mice Exhibited ERS in Brain Tissue

Considering the intimate association between ERS and inflammatory responses [27], as well as the strong association between ERS and neurodegenerative diseases [28], we examined the potential of brain tissue as an indicator of ERS. The results revealed that the ERS signaling pathway was fully activated in the mice at 6 months of age, and this activation persisted until 12 months of age (*p* < 0.05, *p* < 0.01 or *p* < 0.0001) (Figure 6A). Studies have indicated that the PERK-Beclin1-LC3II signaling pathway can modulate ERS to alleviate inflammatory responses [29]. In this study, qPCR analysis revealed that the expression levels of three key factors in this pathway—PERK, Beclin1, and LC3II—were increased in the nucleus of human α-syn transgenic mice at 6 months of age, and this increase continued until 12 months of age (*p* < 0.05, *p* < 0.01 or *p* < 0.001) (Figure 6B). These findings suggest that the PERK-Beclin1-LC3II signaling pathway was activated in these transgenic mice at 6 months of age. It is worth exploring whether nuclear localization in human α-syn transgenic mice can also alleviate ERS by activating the PERK-Beclin1-LC3II signaling pathway.

### 3.7. Apoptosis in the Brain Tissue of Nuclear Localization Human α-Syn Transgenic Mice

As the site of protein processing, the ER is susceptible to damage under stress conditions, leading to the accumulation of misfolded proteins and a characteristic stress response, known as the unfolded protein response (UPR). Chronic ERS can trigger the UPR to promote apoptosis [30]. Therefore, we examined the degree of apoptosis in the brain tissues of both groups of mice. In male human α-syn transgenic mice with nuclear translocation, the expression levels of the apoptosis-related proteins Bax (*p* < 0.01 or *p* < 0.001) (Figure 7A,C) and p-p53 (*p* < 0.05) (Figure 7B,D) were significantly different at 9 months of age, and these differences persisted until 12 months of age. Additionally, using the KEGG database (https://www.kegg.jp/kegg/pathway.html, accessed on 13 March 2024), we identified a cell apoptosis signaling pathway similar to that identified in our study (Figure 7E). Following endogenous apoptosis stimulation in mammalian cells, the BH3 domain of the antiapoptotic protein B-cell lymphoma-2 (Bcl-2) promotes dimerization among Bcl-2 family members, leading to increased expression of the apoptotic protein Bax. This, in turn, facilitates the binding of the apoptotic protein Smac from the mitochondria to X-linked inhibitor of apoptosis protein (XIAP), thereby neutralizing its inhibitory effect on caspase activity and thus promoting apoptosis.

### 3.8. Bioinformatics Analysis Results

To further investigate the impact of nucleus-localized α-syn on mice, we performed Bulk RNA-seq on brain tissues from 12-month-old male mice, with three samples per group. Differentially expressed genes were analyzed via DEGSeq, and the selection criteria for significant differential expression were based on the fold change and q value (corrected *p* value). Genes with |log2Fold change| ≥ 1.5 and q < 0.01 were considered significantly differentially expressed. A total of 1155 upregulated and 568 downregulated genes were identified. qPCR validation was subsequently conducted on 10 selected differentially expressed genes, confirming significant differences in the expression of IL-1α, TNF, PERK, BECLIN, GABA, IL-6α, P53, LC3II, NOS, and SPAG8 between the two groups. Using the DAVID tool, we performed GO enrichment analysis on these differentially expressed genes and generated a bubble chart (Figure 8A). The GO enrichment results indicated that biological process (BP) terms were significantly enriched in cellular biochemical synthesis and nucleotide transcription; cellular component (CC) terms were enriched mainly in nucleosomes, DNA packaging complexes, nuclei, and protein–DNA complexes; and molecular function (MF) terms were enriched primarily in DNA binding and transcription factor activity and binding to organic cyclic compounds. KEGG enrichment analysis revealed several related signaling pathways, including the cell cycle, endocytosis, protein processing in the ER, various neurodegenerative diseases, and ubiquitin-mediated proteolysis (Figure 8B). Notably, the differentially expressed genes obtained from bulk RNA-seq and protein processing in the endoplasmic reticulum pathway identified through KEGG enrichment analysis were consistent with the experimental results presented in the preceding sections.

## 4. Discussion

Research on PD has been ongoing since it was first described in 1817 [31]. Given that the primary pathological hallmark of PD is the presence of Lewy bodies [32] and that these Lewy bodies are composed of α-syn, studying α-syn has become a major focus in PD research.

In recent years, the nuclear localization of α-syn has been increasingly observed in the tissues of patients with various neurodegenerative diseases [33]. However, the relationship between α-syn nuclear localization and PD, which ranks as the second most common neurodegenerative disease, remains unclear. Therefore, in this study, we aimed to investigate this relationship via a transgenic mouse model in which human α-syn was localized to the cell nucleus. To achieve this goal, we selected the EF1A promoter, known for its high stability in embryonic stem cells [34], and the lentiviral vector, which offered good safety and high integration efficiency [35,36]. We constructed a lentiviral vector (pLV-hSNCA-NLS) by linking the nuclear localization signal (NLS) from Simian Vacuolating Virus 40 (SV40) to the cDNA of human SNCA. Additionally, we created a control vector (pLV-EGFP) expressing enhanced green fluorescent protein (EGFP). These vectors were then injected into fertilized mouse eggs to establish a transgenic mouse model with nucleus-localized human α-syn. The successful establishment of this model was subsequently confirmed through PCR, genomic DNA sequencing, Western blotting, and immunofluorescence techniques.

However, one limitation of the current study is the absence of a nuclear export signal group, which may influence the interpretation of our findings. However, when constructing the lentiviral vector, we chose the EF1A promoter due to this stability in embryonic stem cells and its ability to support long-term, robust transgene expression [35]. Thus, EF1A was used to drive α-syn expression. Additionally, α-syn lacks a canonical nuclear-localization signal (NLS); therefore, we fused the well-characterized SV40 NLS to α-syn to achieve specific nuclear import, a strategy employed by several prior studies [9,37]. Our results confirm its efficient nuclear targeting (Figure 2 in manuscript). In our preliminary work we found that nuclear α-syn induced hippocampal DNA damage and consequent cognitive and motor deficits, whereas cytoplasm-localized α-syn produced neither neuroinflammation nor behavioral abnormalities [38]. We further demonstrated that, compared with cytoplasm-localized α-syn, nuclear α-syn reduced hippocampal neurogenesis and evoked anxiety-like behaviors [39]. These prior findings underscore the specific contribution of nuclear α-syn to PD-like motor dysfunction.

Behavioral tests were initiated when the offspring mice reached 1 month of age. The experimental group exhibited significant motor dysfunction as early as 2 months of age, which persisted until 12 months of age. Previous studies have indicated that transgenic mice overexpressing α-syn typically develop pronounced motor dysfunction [13,40,41]. However, with respect to endogenous α-syn accumulation in the nucleus, transgenic mice only exhibit age-dependent motor and gastrointestinal dysfunction at 9 months of age, with less prominent pathological features, such as cortical atrophy in the brain [13]. In contrast, studies using transgenic mice that overexpress only α-syn have shown that these mice develop motor dysfunction at 14 months of age [41]. Even in transgenic mice that exhibit pronounced motor dysfunction earlier, such deficits do not appear until 6 months of age [40]. Compared with these findings, our study, which added a nuclear localization signal to the C-terminus of human α-syn, revealed significant motor dysfunction in mice as early as 2 months of age. These findings suggest that the nuclear localization of α-syn may accelerate the onset of motor dysfunction in mice, thus highlighting the distinct advantages of this model in studying PD-like motor impairments in mice. Anxiety, a typical nonmotor symptom of PD, is present in approximately one-third of PD patients during the course of the disease [24]. In our study, nuclear localization in human α-syn transgenic mice resulted in significant anxiety-like behaviors at 9 months of age, which persisted until 12 months. Moreover, qPCR analysis revealed that the expression levels of the GABA gene, which was associated with emotional regulation, were significantly decreased at both 9 and 12 months of age.

Microglia and astrocytes can participate in inflammatory responses in neurodegenerative diseases by acting as local immune cells [42]. During periods of homeostasis disruption or loss, astrocyte responses to environmental changes include hypertrophic morphology, the upregulation of astrocyte markers (such as GFAP), and the production of inflammatory cytokines [43,44]. In this study, Western blotting revealed that, compared with that in EGFP mice, the nuclear localization of GFAP in human α-syn mice was increased at 2 months of age, and this increase persisted until 12 months of age. These findings indicate that significant inflammatory reactions occurred in the brain tissues of human α-syn transgenic mice with nuclear localization. We further validated this finding by designing qPCR primers for inflammatory factors (IL-1α, IL-1β, IFN-γ, IL-6, NF-κB, and NOS2). The results were consistent with the GFAP findings: significant differences in inflammatory factors were observed in the brain tissues of the two groups of mice starting at 2 months of age, which coincided with the onset of pronounced motor dysfunction in the mice. Therefore, we speculate that the nuclear localization of α-syn may induce central nervous system inflammatory responses in nucleus-localized human α-syn transgenic mice, thereby leading to motor dysfunction in these mice. In recent years, studies have revealed a close relationship between neurodegenerative diseases and ERS [28], and ERS is also closely linked to inflammatory responses [27]. Therefore, in this study, we examined indicators of ERS. The expression levels of the genes encoding key proteins in the three ERS signaling pathways—ATF6, PERK, and IRE1—were significantly elevated in the brain tissues of human α-syn nuclear localization transgenic mice at 6 months of age, suggesting that the ERS pathways might be activated. Does the persistent inflammatory response in the brain tissue of these mice induce the occurrence of ERS? Through which pathway does the inflammatory response induce ERS? To address these questions, we reviewed the literature and reported that activation of the PERK-Beclin1-LC3II pathway, which inhibits the phosphorylation of the upstream ERS—PERK-eIF2α pathway, can ameliorate inflammatory responses [29]. Consequently, through qPCR analysis, we found that at the same time point, namely, at 6 months of age, the expression levels of the autophagy marker protein Beclin1 and the microtubule-associated protein LC3II were also significantly increased. Our study preliminarily suggested that ERS and the PERK-Beclin1-LC3II signaling pathway might be activated simultaneously. Therefore, we speculate that the chronic inflammatory response in the brain tissue of nuclear localization human α-syn transgenic mice can lead to ERS and that PERK-Beclin1-LC3II signaling pathway activation may occur to alleviate this effect. Our future work will focus on the relationship between neuroinflammation and ERS and the underlying molecular mechanisms involved. Under prolonged ERS, the unfolded protein response can promote apoptosis [45]. In this study, Western blot revealed that in the brain tissue of nuclear localization human α-syn transgenic mice, the expression levels of the apoptotic protein Bax and the apoptosis-related protein *p*-p53 significantly increased at 9 months of age. The high expression of the Bax protein persisted until 12 months of age, whereas p-p53 protein expression, although it showed a tendency to remain elevated, did not reach statistical significance at that time point. This finding is corroborated by the apoptosis signaling pathway obtained through the online KEGG database (https://www.kegg.jp/kegg/pathway.html, accessed on 13 March 2024), preliminarily indicating that ERS can promote apoptosis.

Notably, the differentially expressed genes identified through bulk RNA-seq included not only the inflammatory factors IL-1α, IL-6α, and TNF but also key components of the PERK-Beclin1-LC3II signaling pathway, including PERK, BECLIN, and LC3II. Additionally, apoptosis-related genes, including P53 and the emotion-associated gene GABA, were also identified. KEGG enrichment analysis of these differentially expressed genes revealed pathways related to protein processing in the endoplasmic reticulum and various neurodegenerative diseases. These bioinformatics analyses lend further credibility to our findings.

## 5. Conclusions

This study successfully established a transgenic mouse model with nuclear localization human α-syn via a lentiviral vector carrying the SV40-NLS signal. Human α-syn was widely expressed in various mouse tissues and effectively colocalized with the cell nucleus. At 2 months of age, the mice exhibited pronounced astrocyte proliferation, inflammatory responses, and motor dysfunction. Anxiety-like behaviors were also observed as the mice aged. This model effectively simulates PD-like behavioral manifestations in humans, including motor and emotional manifestations. Due to the early onset of motor dysfunction, this transgenic mouse model can serve as a novel animal model for studying PD-related motor impairments and provide foundational data for elucidating the pathogenic mechanisms of human α-syn nuclear localization.

## 6. Patents

The transgenic mouse model presented in this study has been filed as a provisional patent (Application No.202211624399).

## Figures and Tables

**Figure 1 diseases-13-00261-f001:**
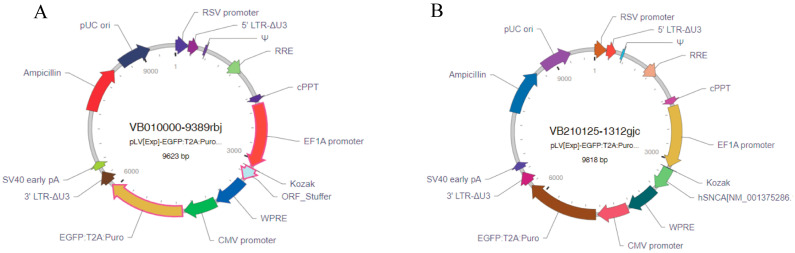
Lentiviral vector construction results. (**A**) EGFP lentiviral vector mapping. (**B**) hSN-CA-NLS lentiviral vector mapping.

**Figure 2 diseases-13-00261-f002:**
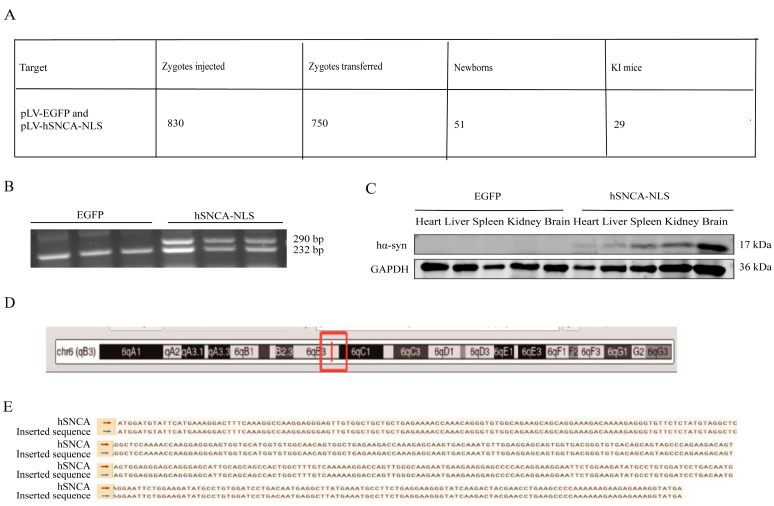
Generation of human α-syn transgenic mice. (**A**) Mouse information summary table. (**B**) PCR genotyping of EGFP and hSNCA-NLS transgenic mice. To confirm the presence of the hα-syn, two primer pairs were designed. Primer pair 1 targets the human α-syn gene, while primer pair 2 is specific to the EGFP element, ensuring amplification occurs only in hα-syn transgenic mice. The expected PCR products are 232 bp and 290 bp, respectively. (**C**) Western blot analysis was performed to detect the expression levels of human α-syn in heart, liver, spleen, kidney, and brain tissues of the transgenic mice. (**D**) The red box indicates the insertion site of the hSNCA-NLS gene on mouse chromosome 6. (**E**) Comparison of whole genome sequencing results of human α-syn transgenic mice. Original images are provided in Appendix A.

**Figure 3 diseases-13-00261-f003:**
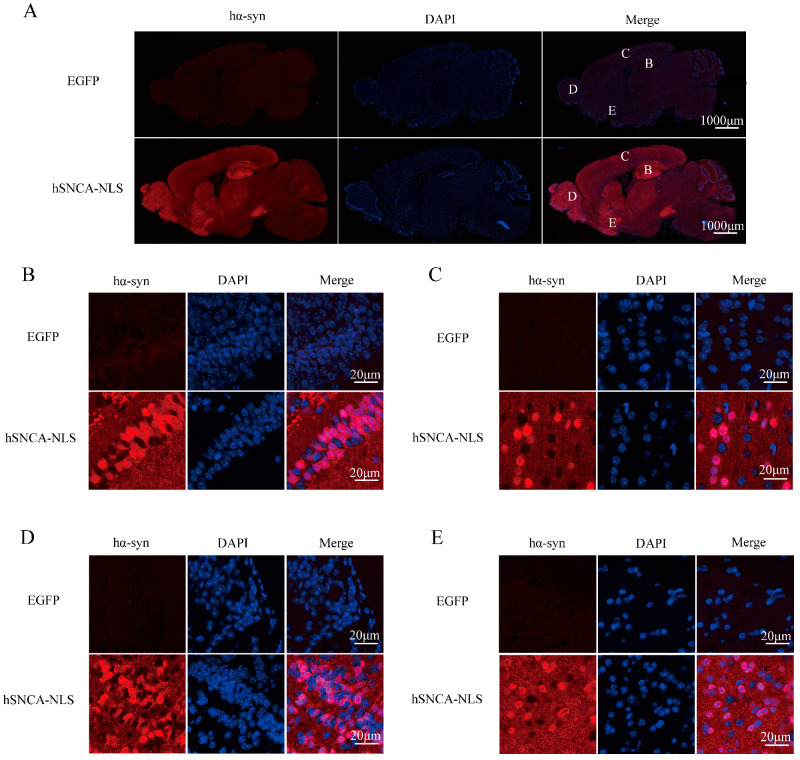
Human α-syn exhibited distinct colocalization with the cell nucleus. (**A**) Expression of human α-syn in the entire brain region. B, C, D, and E represent the hippocampus, cerebral cortex, olfactory bulb, and ventral tegmental area, respectively. (**B**) Colocalization of human α-syn in the hippocampus. (**C**) Colocalization of human α-syn in the cerebral cortex. (**D**) Colocalization of human α-syn in the olfactory bulb. (**E**) Colocalization of human α-syn in the ventral tegmental area. Original images are provided in Appendix A.

**Figure 4 diseases-13-00261-f004:**
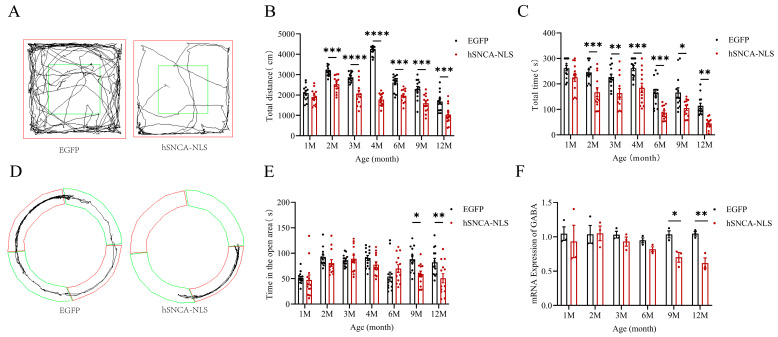
Motor dysfunction and anxiety-like symptoms in nuclear localization human α-syn transgenic mice. (**A**) Trajectories of EGFP mice and hSNCA-NLS transgenic mice in open field test, The central area (green box) are used to assess exploratory behavior and anxiety levels. (**B**) Comparison of distance traveled in the open field test at 1, 2, 3, 4, 6, 9, and 12 months of age (*n* = 13 per group). (**C**) Comparison of the residence time in the rotarod test at 1, 2, 3, 4, 6, 9, and 12 months of age (*n* = 13 per group). (**D**) Trajectories of EGFP mice and hSNCA-NLS transgenic mice in the O maze test, The open area (green ring) and the closed area (red ring) are used to assess anxiety-like behavior. (**E**) Comparison of distance traveled in open areas of the O maze in 1, 2, 3, 4, 6, 9, and 12 months of age (*n* = 13 per group). (**F**) Real-time PCR was performed to assess GABA gene expression in hSNCA-NLS transgenic mice (*n* = 3 per group). * *p* < 0.05, ** *p* < 0.01, *** *p* < 0.001, and **** *p* < 0.0001.

**Figure 5 diseases-13-00261-f005:**
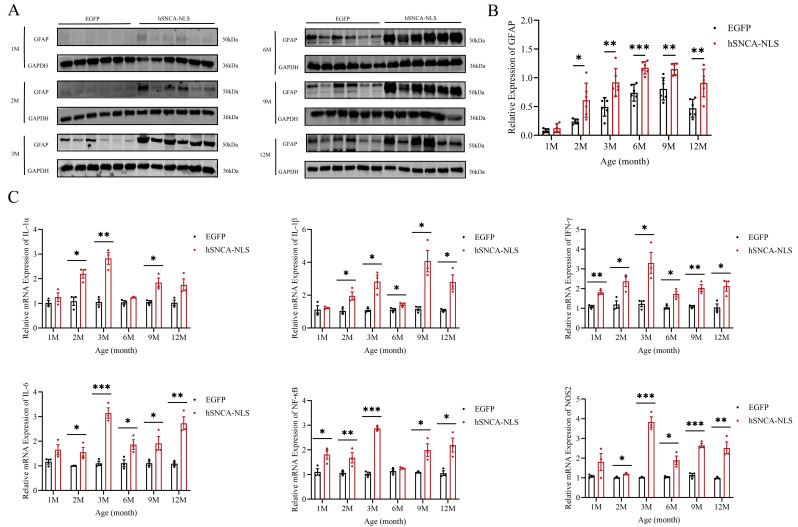
Astrocyte proliferation and inflammatory response in brain tissues of nuclear localization human α-syn transgenic mice. (**A**) Western blot analysis was performed to detect the expression of GFAP at 1, 2, 3, 6, 9, and 12 months of age. (**B**) Statistical graph of WB results for GFAP (*n* = 6 per group). (**C**) Real-time PCR was performed to assess the expression levels of inflammatory cytokines at 1, 2, 3, 6, 9, and 12 months of age (*n* = 6 per group). * *p* < 0.05, ** *p* < 0.01, *** *p* < 0.001. Original images are provided in Appendix A.

**Figure 6 diseases-13-00261-f006:**
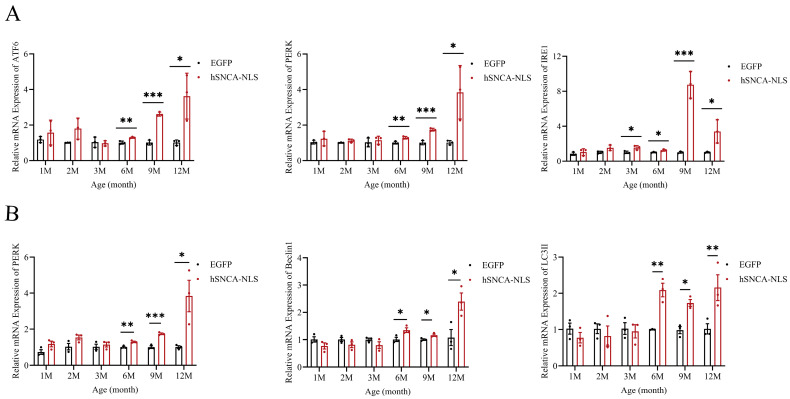
Endoplasmic reticulum stress occurs in brain tissues of nuclear localization human α-syn transgenic mice. (**A**) Real-time PCR was performed to assess the expression levels of ATF6, PERK, and IRE1 at 1, 2, 3, 6, 9, and 12 months of age (*n* = 3 per group). (**B**) Real-time PCR was performed to assess the expression levels of PERK, Beclin1, and LC3II at 1, 2, 3, 6, 9, and 12 months of age (*n* = 3 per group). * *p* < 0.05, ** *p* < 0.01, *** *p* < 0.001.

**Figure 7 diseases-13-00261-f007:**
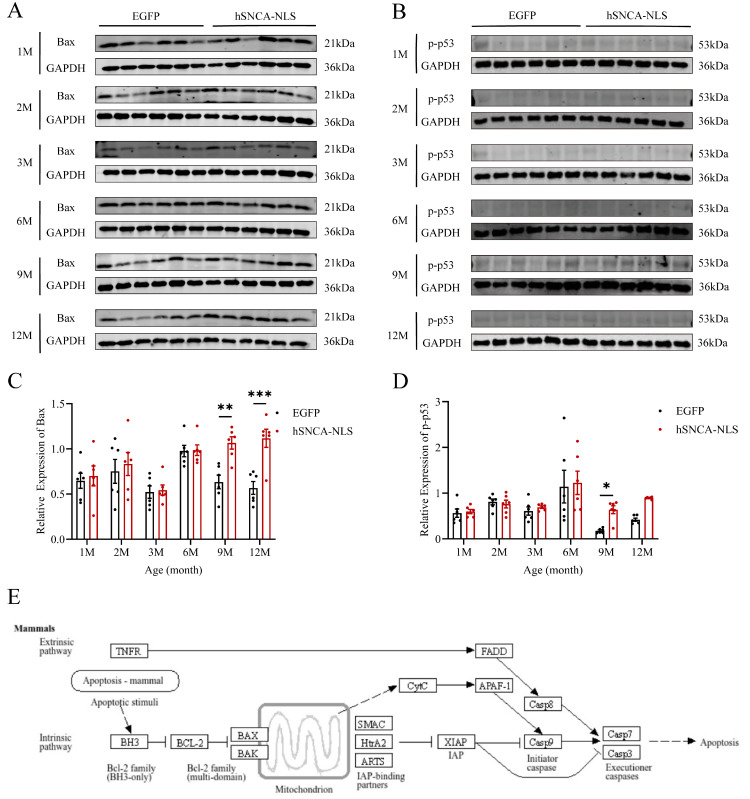
Apoptosis in brain tissue of nuclear localization human α-syn transgenic mice. (**A**) Western blot analysis was performed to detect the expression levels of Bax. (**B**) Western blot analysis was performed to detect the expression levels of p-p53. (**C**) Statistical graph of WB results for Bax (*n* = 6 per group). (**D**) Statistical graph of WB results for p-p53 (*n* = 6 per group). (**E**) Apoptosis signaling pathway diagram. * *p* < 0.05, ** *p* < 0.01, and *** *p* < 0.001. Original images are provided in Appendix A.

**Figure 8 diseases-13-00261-f008:**
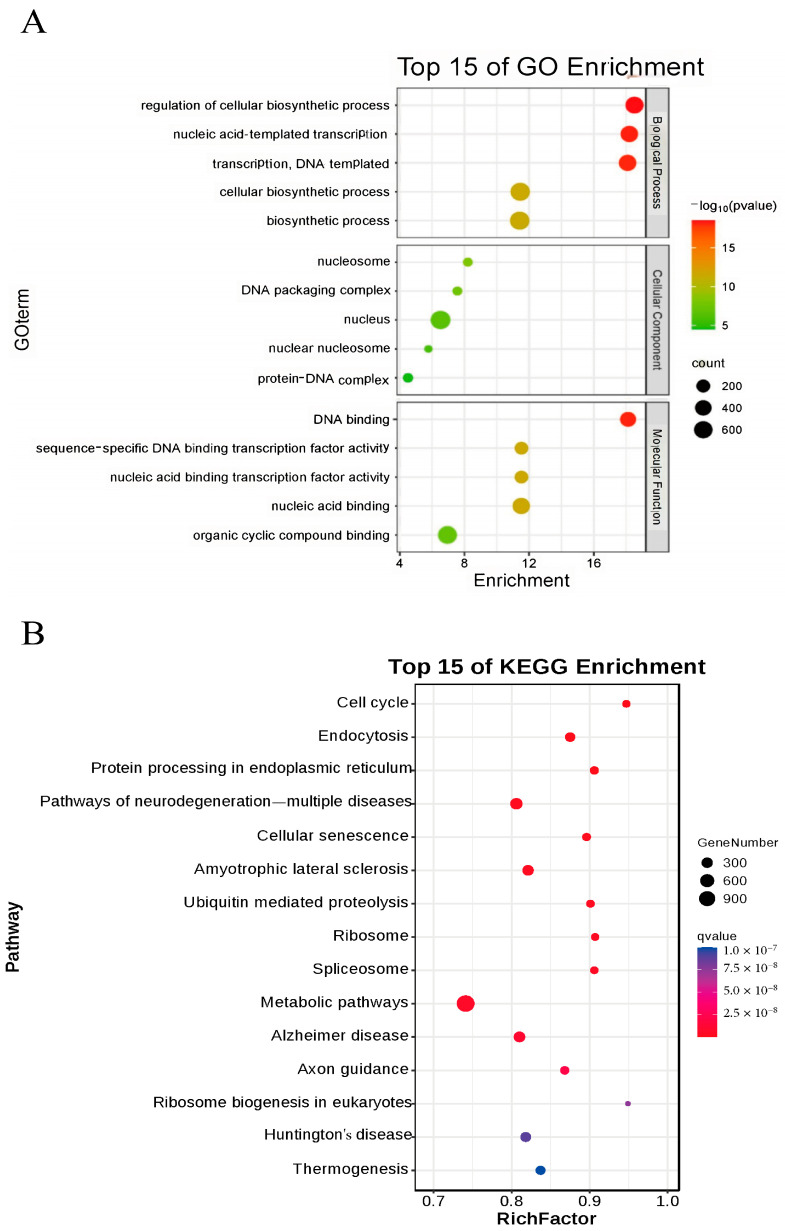
Bioinformatics analysis results. (**A**) GO analysis of differentially expressed genes in nuclear localization human α-syn transgenic mice. (**B**) Top 15 pathways enriched in KEGG-analyzed nuclear localization human α-syn transgenic mice.

## Data Availability

The data supporting the conclusions of this article are included within the article or are available from the authors upon reasonable request.

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
