# Peer review of "Establishment and Characterization of Behavioral Changes in the Nuclear Localization Human α-Synuclein Transgenic Mice"

_diseases, 2025, doi:10.3390/diseases13080261_

Round 1
Reviewer 1 Report
Comments and Suggestions for Authors
In this study, the authors demonstrated the effect of nuclear-localized human alpha-synuclein on neuroinflammation in mice. The main manuscript is well-written. It is also logical. And it is narrated well. However, the design of the experiments is lacking in some areas.
The authors set up the EGFP group as a control group. However, it is not a valid control for NLS-tagged human alpha-synuclein.
If the authors intend to demonstrate a difference in effect with overexpressed alpha-synuclein in the nucleus, they should firstly compare it to NLS-tagged EGFP overexpression. Secondly, if they intend to verify that nuclear alpha-synuclein causes neuroinflammation, they should compare it to the overexpression of human alpha-synuclein with a NES-tag that does not allow it to enter the nucleus.
Author Response
Comments 1: The authors set up the EGFP group as a control group. However, it is not a valid control for NLS-tagged human alpha-synuclein. If the authors intend to demonstrate a difference in effect with overexpressed alpha-synuclein in the nucleus, they should firstly compare it to NLS-tagged EGFP overexpression. Secondly, if they intend to verify that nuclear alpha-synuclein causes neuroinflammation, they should compare it to the overexpression of human alpha-synuclein with a NES-tag that does not allow it to enter the nucleus.
Response 1: We are very grateful for your guidance. As you rightly observed, our experimental design lacked a group with human alpha-synuclein with a NES-tag, which is indeed a significant limitation. We will incorporate this group in future work. Meanwhile, we have added a paragraph (paragraph 3, line396-409) to the Discussion that addresses the potential consequences of this omission. The revised text reads as follows:
“However, one limitation of the current study is the absence of a nuclear export signal (NES) group, which may influence the interpretation of our findings. However, when constructing the lentiviral vector, we chose the EF1A promoter due to this stability in embryonic stem cells and its ability to support long-term, robust transgene expression [1]. Thus, EF1A was used to drive α-syn expression. Additionally, α-syn lacks a canonical nuclear-localization signal (NLS); therefore, we fused the well-characterized SV40 NLS to α-syn to achieve specific nuclear import, a strategy employed by several prior studies [2,3]. Our results confirm efficient nuclear targeting (Figure 2 in manuscript).In our preliminary work we found that nuclear α-syn induced hippocampal DNA damage and consequent cognitive and motor deficits, whereas cytoplasm-localized α-syn produced neither neuroinflammation nor behavioral abnormalities [4]. We further demonstrated that, compared with cytoplasm-localized α-syn, nuclear α-syn reduced hippocampal neurogenesis and evoked anxiety-like behaviors [5]. These prior findings underscore the specific contribution of nuclear α-syn to PD-like motor dysfunction.”
For these reasons, we did not include a hSNCA-NES group in the present study and used only pLV-EGFP as a control. We recognize that this constitutes a design flaw and will address it in subsequent experiments. Thank you again for your invaluable feedback.
- Moreira, A.S.; Cavaco, D.G.; Faria, T.Q.; Alves, P.M.; Carrondo, M.J.T.; Peixoto, C., Advances in lentivirus purification. Biotechnology Journal 2020, 16.
- Rousseaux, M.W.C.; de Haro, M.; Lasagna-Reeves, C.A.; De Maio, A.; Park, J.; Jafar-Nejad, P.; Al-Ramahi, I.; Sharma, A.; See, L.; Lu, N., et al., Trim28 regulates the nuclear accumulation and toxicity of both alpha-synuclein and tau. eLife 2016, 5.
- Khairnar, A.; Ruda-Kucerova, J.; Szabó, N.; Drazanova, E.; Arab, A.; Hutter-Paier, B.; Neddens, J.; Latta, P.; Starcuk, Z.; Rektorova, I., Early and progressive microstructural brain changes in mice overexpressing human α-synuclein detected by diffusion kurtosis imaging. Brain, Behavior, and Immunity 2017, 61, 197-208.
- Pan, Y.; Zong, Q.; Li, G.; Wu, Z.; Du, T.; Huang, Z.; Zhang, Y.; Ma, K., Nuclear localization of alpha-synuclein affects the cognitive and motor behavior of mice by inducing DNA damage and abnormal cell cycle of hippocampal neurons. Frontiers in Molecular Neuroscience 2022, 15.
- Pan, Y.; Zong, Q.; Li, G.; Wu, Z.; Du, T.; Zhang, Y.; Huang, Z.; Ma, K., Nuclear localization of alpha-synuclein induces anxiety-like behavior in mice by decreasing hippocampal neurogenesis and pathologically affe cting amygdala circuits. Neuroscience letters 816, 137490.
We appreciate for your warm work earnestly, and hope that the correction will meet with approval. We have made revision which highlighted in yellow in the paper.
Reviewer 2 Report
Comments and Suggestions for Authors
The manuscript by Ziou Wang et al., entitled “Establishment and Characterization of Behavioral Changes in the Nuclear Localization Human α-Synuclein Transgenic Mice”, submitted to Diseases, presents a timely and methodologically rigorous investigation into the pathological role of nuclear-localized α-synuclein in Parkinson’s disease (PD). By generating a novel transgenic mouse model that specifically expresses human α-synuclein with a nuclear localization signal, the authors effectively recapitulate key features of PD, including early-onset motor deficits, anxiety-like behaviors, neuroinflammation, endoplasmic reticulum stress, and neuronal apoptosis. The study impressively integrates behavioral phenotyping, molecular analyses, and single-cell transcriptomics to elucidate the mechanistic links between nuclear α-synuclein and neurodegeneration. This comprehensive approach makes the manuscript a valuable contribution to the field of synucleinopathies and supports its suitability for publication in Diseases; however, several concerns need to be addressed prior to publication
Major comments:
- While the behavioral and molecular analyses suggest neurodegenerative changes, the manuscript lacks direct histological evidence of neuronal loss. There are no data from Nissl staining, NeuN immunohistochemistry, or tyrosine hydroxylase (TH) staining in the substantia nigra, markers that are essential for confirming dopaminergic neurodegeneration and neuronal depletion. This represents a major limitation when claiming the model recapitulates Parkinson’s disease-like pathology.
- Although the authors infer relevance to PD pathology, the manuscript lacks direct evidence of Lewy body formation. Specifically, no aggregation markers such as phosphorylated α-syn at Ser129, Thioflavin S staining, or ubiquitin-positive inclusions were assessed. These are essential for validating the presence of α-syn aggregates and are considered key hallmarks of Lewy body pathology. This omission limits the strength of the claim that the model recapitulates PD-related neuropathology.
- The manuscript does not specify the sex of the animals used in behavioral or molecular experiments. This is a critical omission, as PD exhibits known sex-related differences in incidence, symptomatology, and neuroinflammatory responses. The absence of sex-based stratification or reporting could overlook important biological variability and limits the generalizability of the findings. The authors should clarify the sex of the animals used and, where possible, consider reanalyzing the data with sex as a biological variable.
- The experimental design lacks a critical control group: transgenic mice overexpressing wild-type (cytoplasmic) human α-syn without the nuclear localization signal (NLS). While EGFP-only mice serve as a general negative control, they do not allow for distinguishing effects specifically attributable to nuclear localization versus those arising from α-syn in overexpression itself. Given that overexpression of cytoplasmic α-syn is known to cause neuroinflammation and motor deficits in other models, the inclusion of a wild-type α-syn group would have strengthened the study’s conclusions. At minimum, the authors should discuss this limitation and its potential impact on data interpretation.
- While single-cell RNA sequencing (scRNA-seq) was performed, the results are presented only in aggregate without cell-type-specific resolution. The lack of analysis distinguishing changes across neurons, astrocytes, microglia, or other brain cell types weakens the ability to interpret which populations contribute to the observed phenotypes. For example, it remains unclear whether the inflammatory response is primarily astrocyte- or microglia-derived. Given that scRNA-seq allows such resolution, the authors should consider including a cell-type-specific analysis or, alternatively, acknowledge and discuss this limitation in the manuscript.
- In Figure 3, the sagittal brain sections shown for different regions (e.g., hippocampus, cortex, olfactory bulb) appear to be taken from different anatomical planes or levels. It would improve clarity if the authors could confirm whether all images were acquired from matched sagittal sections or, if not, justify the selection of different planes. Using consistent anatomical levels would strengthen the anatomical interpretation and comparability of regional α-syn localization.
- In Figure 5C, it would improve clarity and interpretability if the authors presented each inflammatory cytokine’s expression level across timepoints individually, similar to the approach used for GFAP in Figure 5B. This would allow a clearer comparison of how each cytokine changes with age and may help elucidate which inflammatory mediators show the earliest or most pronounced upregulation. If this was not feasible due to data limitations, a brief explanation in the figure legend or discussion would be helpful.
- A similar issue applies to Figure 6 as noted for Figure 5C. Presenting each ER stress-related gene (e.g., ATF6, PERK, IRE1) in separate time-course plots, rather than grouping all genes together per timepoint, would make it easier to interpret the temporal dynamics of each gene’s expression. This would provide a clearer view of how individual stress markers change with age in the transgenic mice and strengthen the analysis of progressive ER stress. If the authors chose not to display the data this way for specific reasons, it would be helpful to mention this in the figure legend or discussion.
- In Figure 7D, the elevation of p-p53 expression appears clearly significant at 9 months but is much less pronounced at 12 months. However, in the discussion (lines 453–454), the authors state that “the p-p53 protein expression also tended to remain elevated between months 9–12,” which may be an overstatement if the difference at 12 months is not statistically significant. The authors are encouraged to either clarify the statistical significance of this observation or revise the statement to more accurately reflect the data shown.
Minor comments:
- Line 30: It is recommended to avoid starting a sentence with an abbreviation. Consider revising “PD represents...” to “Parkinson’s disease (PD) represents...” for improved readability and formality.
- Figure 2: The authors are advised to specify in the main text that three mice from each group were used to validate transgene expression via PCR and western blotting, as shown in the figure. This would clarify sample size and improve methodological transparency.
- Figure 4: The legend should explicitly mention that GABA gene expression was assessed by qPCR, as currently this information is only found in the discussion (lines 411–412). Including it in the figure legend will aid clarity for readers interpreting the data.
- Abbreviations: The entry for “BCl2” should be corrected to match the form used in the main text, i.e., “Bcl-2,” for consistency and accuracy.
Round 2
Reviewer 1 Report
Comments and Suggestions for Authors
The authors have addressed the issuse that I ponited out.
Reviewer 2 Report
Comments and Suggestions for Authors
I appreciate the effort made to address most of the comments raised in the initial review. While some points remain unaddressed, the authors' acknowledgment of these as current limitations and their plan to address them in future studies is noted. The manuscript presents valuable findings and is generally well-structured. I recommend the manuscript for acceptance.